# Chitosan–Polyethylene Glycol Inspired Polyelectrolyte Complex Hydrogel Templates Favoring NEO-Tissue Formation for Cardiac Tissue Engineering

**DOI:** 10.3390/gels10010046

**Published:** 2024-01-08

**Authors:** Angelo Keklikian, Natan Roberto de Barros, Ahmad Rashad, Yiqing Chen, Jinrui Tan, Ruoyu Sheng, Dongwei Sun, Huinan Liu, Finosh G. Thankam

**Affiliations:** 1Department of Translational Research, College of Osteopathic Medicine of the Pacific and Graduate College of Biomedical Sciences, Western University of Health Sciences, Pomona, CA 91766, USA; akeklo28@gmail.com; 2Terasaki Institute for Biomedical Innovation (TIBI), Los Angeles, CA 90064, USA; nbarros@terasaki.org (N.R.d.B.); ahmad.elsebahy@terasaki.org (A.R.); 3Department of Bioengineering, University of California, Riverside, CA 92521, USA; yiqing.chen@ucr.edu (Y.C.); ruoyu.sheng@email.ucr.edu (R.S.); dongwei.sun@email.ucr.edu (D.S.); huinanliu@engr.ucr.edu (H.L.)

**Keywords:** neo-tissue formation, cardiac tissue engineering, polyelectrolyte hydrogels, chitosan, regenerative cardiology

## Abstract

Neo-tissue formation and host tissue regeneration determine the success of cardiac tissue engineering where functional hydrogel scaffolds act as cardiac (extracellular matrix) ECM mimic. Translationally, the hydrogel templates promoting neo-cardiac tissue formation are currently limited; however, they are highly demanding in cardiac tissue engineering. The current study focused on the development of a panel of four chitosan-based polyelectrolyte hydrogels as cardiac scaffolds facilitating neo-cardiac tissue formation to promote cardiac regeneration. Chitosan-PEG (CP), gelatin-chitosan-PEG (GCP), hyaluronic acid-chitosan-PEG (HACP), and combined CP (CoCP) polyelectrolyte hydrogels were engineered by solvent casting and assessed for physiochemical, thermal, electrical, biodegradable, mechanical, and biological properties. The CP, GCP, HACP, and CoCP hydrogels exhibited excellent porosity (4.24 ± 0.18, 13.089 ± 1.13, 12.53 ± 1.30 and 15.88 ± 1.10 for CP, GCP, HACP and CoCP, respectively), water profile, mechanical strength, and amphiphilicity suitable for cardiac tissue engineering. The hydrogels were hemocompatible as evident from the negligible hemolysis and RBC aggregation and increased adsorption of plasma albumin. The hydrogels were cytocompatible as evident from the increased viability by MTT (>94% for all the four hydrogels) assay and direct contact assay. Also, the hydrogels supported the adhesion, growth, spreading, and proliferation of H9c2 cells as unveiled by rhodamine staining. The hydrogels promoted neo-tissue formation that was proven using rat and swine myocardial tissue explant culture. Compared to GCP and CoCP, CP and HACP were superior owing to the cell viability, hemocompatibility, and conductance, resulting in the highest degree of cytoskeletal organization and neo-tissue formation. The physiochemical and biological performance of these hydrogels supported neo-cardiac tissue formation. Overall, the CP, GCP, HACP, and CoCP hydrogel systems promise novel translational opportunities in regenerative cardiology.

## 1. Introduction

Neo-tissue formation and host tissue regeneration are the major checkpoints in determining the success of cardiac tissue engineering which is driven by the chemistry and functional performance of scaffolds employed. This process is decided by the biocompatibility, physiochemical features and the contact guidance provided by the hydrogel scaffolds for promoting the proliferation and signaling by regenerative cell phenotypes [1]. Hence, neo-tissue formation is driven by the cardiac-mimetic features of the hydrogel templates employed, allowing for the seeded/host cell migration and accelerating the deposition of the ECM components and angiogenesis [2,3,4]. Unfortunately, despite the greater demand, the capability of hydrogel templates in promoting neo-cardiac tissue formation is currently limited. Interestingly, polyelectrolyte complex hydrogels promise tissue engineering applications owing to their tunable features, electrical conductivity and structural similarities with mammalian ECM [5]. However, to the best of our knowledge, the polyelectrolyte complex hydrogels that supports neo-cardiac tissue formation is currently unavailable.

Chitosan-based hydrogels demonstrates myocardial regeneration as evidenced by the improved myocardial wall integrity, decreased infarct expansion and left ventricular (LV) remodeling, and retention of contractile function in pre-clinical myocardial infarction (MI) models [6,7,8]. Additionally, the inherent antimicrobial and antioxidant properties and the ability to form spontaneous polyelectrolyte complex impart additional benefits to chitosan for cardiac applications [9]. Likewise, hydrogels based on polyethylene glycol (PEG), a FDA approved synthetic polymer, have been praised for improving the cardiac function in MI models owing to their exceptional biocompatibility and mechanical strength [10,11]. Additionally, the ECM-derived protein gelatin and cardiac ECM component hyaluronic acid (HA) offer significant benefits for cardiac tissue engineering by providing biological adhesion sites and homely environment to the seeded cells and the subsequent neo-tissue formation [12,13,14,15]. Interestingly, the ample positive charge density in chitosan and the anionic functional groups in gelatin, HA and PEG provides enhanced opportunities for forming polyelectrolyte complex hydrogels. However, such templates supporting neo-tissue formation for cardiac applications are currently unavailable. Given this background, we hypothesize that chitosan-based polyelectrolyte complex hydrogels containing the native cardiac ECM biomolecules gelatin and HA promote neo-cardiac tissue formation. Hence, the current study aims to synthesize novel polyelectrolyte hydrogels from chitosan, PEG, gelatin, and HA, characterize the properties and performance, and evaluate their potential in neo-tissue formation for cardiac tissue engineering applications. The hydrogels were tested for physiochemical, mechanical and thermal characteristics and biological performance to assess their ability for neo-tissue formation.

## 2. Results and Discussion

CP blend in the aqueous phase was casted, allowing for the formation of the parent polyelectrolyte hybrid CP hydrogel. The CP hydrogels were interpenetrated with the biomolecules, gelatin and HA for improved biocompatibility and biological performance. The addition of gelatin to the CP blend resulted in GCP hydrogel. Similarly, CP was blended with hyaluronic acid to form HACP hydrogel. CoCP hydrogel was synthesized by incorporating both gelatin and HA to the CP blend. All four preparations, CO, GCP, HACP and CoCP, were casted at 55 °C to prevent the denaturation of biological domains in the gelatin and HA components of the hydrogels, thereby ensuring the maximum biological responses. Also, the four hydrogels exhibited flexibility of handling using tweezers without breaking.

### 2.1. Physiochemical Characterization

#### 2.1.1. ATR-IR Spectroscopy

AT-IR spectroscopy was used to determine the surface functional groups on CP, GCP, HACP, and CoCP hydrogels (Figure 1A–D). A broad peak at about 3200 cm^−1^ denoted the presence of -OH groups with a peak area of 5281.1 ± 1427.7 for CP, 4479.6 ± 1547.6 for GCP, 3664.9 ± 1010.4 for HACP, and 4492.2 ± 639.2 for CoCP. The difference in the -OH peak intensity was statistically not significant among the groups, thereby suggesting similar level of surface -OH groups contributed by chitosan, PEG, gelatin, and HA. C-H stretch was demonstrated by a peak around 2800 cm^−1^ with a peak area of 1142.7 ± 291.9 for CP, 1338.2 ± 26.7 for GCP, 2537.3 ± 557.8 for HACP, and 1847.2 ± 378.0 for CoCP. HACP displayed significantly higher intensity for the C-H stretch when compared to GCP (*p* = 0.0062) and CoCP (*p* = 0.0007). However, the difference in the C-H peak intensity was statistically not significant among the other groups, thus suggesting similar surface chemistry for C-H. The peak around 1550 cm^−1^ demonstrated N-H bending with a peak area of 618.3 ± 281.4 for CP, 527.2 ± 322.6 for GCP, 365.1 ± 182.5 for HACP, and 608.8 ± 134.2 for CoCP, thereby suggesting the formation of polyelectrolyte complex [16]. Also, the difference in the N-H bending peak intensity was statistically not significant among the groups suggesting similar surface chemistry for N-H, revealing the presence of chitosan, gelatin, and HA. C-H bending of methyl groups was characterized by a peak at about 1440 cm^−1^ with a peak area of 294.7 ± 37.8 for CP, 1264.0 ± 79.3 for GCP, 386.3 ± 54.0 for HACP, and 204.6 ± 32.3 for CoCP. Interestingly, GCP displayed significantly increased C-H bending of methyl groups relative to CP (*p* = 0.0006), HACP (*p* = 0.0023), and CoCP (*p* = 0.0003). However, the difference in the C-H bending peak intensity was statistically not significant among the other groups, thus suggesting similar surface chemistry for C-H. Lastly, C-O stretching of vinyl ethers was characterized by a peak around 1400 cm^−1^ with a peak area of 262.2 ± 163.6 for CP, 192.2 ± 110.2 for GCP, 308.7 ± 129.1 for HACP, and 473.7 ± 238.8 for CoCP. Also, the difference in the C-O stretching peak intensity was statistically not significant among the groups, thereby suggesting the presence of chitosan and PEG fragments on the surface of all the four hydrogels. Overall, the IR spectrum demonstrated similar surface chemistry in all four hydrogels (Figure 1A–D).

#### 2.1.2. Contact Angle

The ACA and RCA for CP hydrogels were 29.38 ± 2.57° and 31.83 ± 2.42°, respectively. GCP displayed an ACA of 43.70 ± 4.60° and 46.55 ± 5.99°. HACP hydrogels exhibited 44.40 ± 3.90° and 42.38 ± 7.13° for ACA and RCA, respectively. ACA for CoCP hydrogels was 47.21 ± 7.20°, and RCA was 49.05 ± 5.39 °. Overall, the contact angle measurements of all the four hydrogels were in the amphiphilic range. Furthermore, the difference between ACA and RCA of CP (*p* = 0.1594), GCP (*p* = 0.4233), HACP (*p* = 0.6359), and CoCP (*p* = 0.6294) hydrogels were not statistically significant, thus suggesting the absence of phase transitions and migration of functional moieties in the hydrogels upon interacting with the aqueous environment (Table 1).

#### 2.1.3. Water Profile

The EWC for CP, GCP, HACP, and CoCP was 70.92 ± 0.94%, 79.12 ± 2.43%, 79.56 ± 3.30, and 80.65 ± 4.54%, respectively. Similarly, the % swelling was 247.3 ± 11.8%, 447.8 ± 65.62%, 551.2 ± 113.6%, and 926.8 ± 245.9% for CP, GCP, HACP, and CoCP, respectively. The values for E and S are displayed in Table 1, which were used to calculate the diffusional exponent (n). The respective diffusional exponent and diffusion constant (k) were 0.0913 and 0.3542 for CP, −0.079 and 0.8751 for GCP, −0.007 and 0.8475 for HACP, and −0.017 and 1.0641 for CoCP. Linear regressions for log(S/E) vs. log(t) are displayed in Figure 1E. The alterations in E, EWC, S and %S were statistically not significant, thereby suggesting similar water profile for CP, GCP, HACP, and CoCP hydrogels. Also, the water profiling displayed the superabsorbent nature of GCP, HACP, and CoCP hydrogels. GCP (*p* < 0.0001), HACP (*p* < 0.0001), and CoCP (*p* < 0.0001) hydrogels displayed significantly increased TWAS compared to the CP hydrogels; however, all the hydrogels exhibited a TWAS in the order of 10^21^ (Table 1).

#### 2.1.4. Cross-Sectional Pore Morphometry

SEM analysis revealed the cross-sectional morphometry of CP, GCP, HACP, and CoCP hydrogels, showing excellent porosity with interconnectivity (Figure 2A–D). Moreover, the pore sizes for the hydrogels were heterogenous and the CP hydrogels displayed significantly decreased pore lengths compared to GCP (*p* < 0.0001), HACP (*p* < 0.0001), and CoCP (*p* < 0.0001) (Table 1) (Figure 2E). Similarly, CP displayed a significantly lower aspect ratio than GCP (*p* = 0.0023) and HACP (*p* = 0.0036); however, the overall trend was toward a oval pore shape (Table 1) (Figure 2F).

#### 2.1.5. Water Transition Status

The DSC thermograms for CP, GCP, HACP and CoCP hydrogels displayed corresponding endothermic and exothermic peaks, reflecting the melting of frozen water and crystallization of freezing water, respectively (Figure 2G–J). CP hydrogel contained 69.58% freezable water and 1.34% non-freezing bound water; GCP hydrogel contained 34.28% freezable water and 44.91% non-freezable water; HACP contained 52.93% freezable water and 26.63% non-freezable water; and CoCP hydrogel contained 71.92% freezable water and 8.74 non-freezing bound water (Table 1) (Figure 2G–J).

### 2.2. Release Profile

CP, GCP, HACP, and CoCP hydrogels demonstrated the ability to absorb and release hydrophobic and hydrophilic dyes. Overall, the release profiles for CP, GCP, HACP, and CoCP displayed a constant release rate of hydrophobic and hydrophilic dyes (Figure 2K,L). Specifically, GCP, HACP, and CoCP hydrogels displayed consistent release profile compared to CP hydrogel.

### 2.3. Conductance

CP hydrogels exhibited a conductance of 0.28 ± 0.01 µS/cm, while the conductance for GCP hydrogels was 0.69 ± 0.07 µS/cm, HACP hydrogels was 2.60 ± 1.23 µS/cm, and CoCP hydrogels was 0.44 ± 0.02 µS/cm (Table 1). HACP displayed significantly increased conductance compared to CP (*p* = 0.0001), GCP (*p* = 0.0008), and CoCP (*p* = 0.0002).

### 2.4. Biodegradation

The CP, GCP, HACP, and CoCP hydrogels were biodegradable, as evidenced by the progressive weight loss upon aging in PBS. The degradation profiles of CP, GCP, HACP, and CoCP hydrogels proceeded as surface erosion in a regulated manner (Figure 2M). Also, the slight drop in pH for CP (−0.03 ± 0.01), GCP (−0.03 ± 0.01), HACP (−0.01 ± 0.00), and CoCP (−0.04 ± 0.01) hydrogels were evident, which suggest very mild acidic degradation products.

### 2.5. Mechanical Characterization

Using UTM, Young’s modulus, maximum tensile stress, and load at failure were calculated for the CP, GCP, HACP, and CoCP hydrogels. CP hydrogels exhibited 1141.0 ± 24.1 kPa Young’s modulus, and 401.9 ± 54.2 kPa tensile stress with 5.62 ± 0.75 N load at failure. Similarly, GCP displayed 179.4 ± 51.5 kPa Young’s modulus, 39.22 ± 2.12 kPa tensile stress, and 0.94 ± 0.03 N load at the failure. The Young’s modulus for HACP hydrogel was 368.4 ± 95.4 kPa, tensile stress was 96.41 ± 18.81 kPa, and load at the failure was 1.74 ± 0.31 N. For CoCP hydrogel, Young’s modulus was 131.7 ± 6.7 kPa, tensile stress was 47.22 ± 5.71 kPa, and load at the failure was 1.34 ± 0.12 N (Table 1). The variations in modulus, tensile stress and load at the failure were statistically not significant among CP, GCP, HACP, and CoCP hydrogels.

### 2.6. Cytocompatibility

Direct contact assay using CP, GCP, HACP, and CoCP hydrogels revealed the absence of changes in the characteristic morphology of H9c2, demonstrating that all four hydrogels were non-cytotoxic and supported the cell proliferation upon contact (Figure 3A–E). Direct contact viability assay displayed >70% viability of H9c2 cells where the respective viability was 71.88 ± 3.26 (*p* < 0.0001), 82.84 ± 2.82 (*p* = 0.0002), 92.44 ± 0.79 (*p* = 0.1215) and 74.09 ± 1.43 (*p* < 0.0001) for CP, GCP, HACP, and CoCP hydrogels (Table 1). Also, the viability assay using the hydrogel extracts for CP hydrogel was 96.1 ± 5.2%, for GCP was 98.8 ± 10.4%, for HACP was 114.4 ± 11.1%, and for CoCP was 94.9 ± 3.2% (Table 1) (Figure 3F). The alteration in cell viability on extracts was statistically not significant among the four hydrogels and the control.

### 2.7. Hemocompatibility

Hemolysis was 0.66 ± 0.02%, 0.76 ± 0.02%, 0.27 ± 0.08%, and 0.63 ± 0.04% for CP, GCP, HACP, and CoCP hydrogels, respectively (Table 1) (Figure 3G). HACP hydrogel exhibited significantly decreased levels of hemolytic potential compared to CP (*p* = 0.0046), GCP (*p* = 0.0004), and CoCP (*p* = 0.0113) hydrogels. Also, the CP, GCP, HACP, and CoCP hydrogels evoked minimal RBC aggregations, rouleaux formations, and changes in characteristic RBC morphology, thus revealing their hemocompatibility (Figure 3H–L). The total serum protein adsorption was 1.67 ± 0.51 µg/µL, 5.48 ± 0.26 µg/µL, 2.78 ± 0.26 µg/µL, 8.19 ± 1.80 µg/µL for CP, GCP, HACP, and CoCP hydrogels, respectively (Table 1) (Figure 3M). CoCP displayed significantly increased adsorption of plasma proteins compared to the control (*p* = 0.0060), CP (*p* = 0.0062) and HACP (*p* = 0.0031); however, the increase was statistically not significant for GCP (*p* = 0.2677). SDS-PAGE analysis revealed that albumin adsorption was prevalent in all the four hydrogels compared to the control (Figure 3N). Alterations in the relative amounts of protein and albumin adsorption were similar among the four hydrogel groups, suggesting similar protein and albumin adsorption capacities.

### 2.8. Biological Performance

The CP, GCP, HACP, and CoCP hydrogels facilitated H9c2 cardiomyoblasts adhesion to the porous network, migration within interstices, and proliferation as observed from the rhodamine staining. The H9c2 cells formed 3D clusters within the hydrogels, highlighting their cytocompatibility and ability to support the 3D growth of cells (Figure 4) (Appendix A). Neo-tissue formation was confirmed using the explant culture from the left ventricle of rat and swine models by growing onto the CP, GCP, HACP, and CoCP hydrogels after seven days. Rhodamine phalloidin staining displayed cytoskeletal organization and neo-tissue formation in both rat (Figure 5) (Appendix A) and swine (Figure 6) (Appendix A) cultures where the neo-tissue adhered to the hydrogels. The cells migrated onto the hydrogel network and into the interstices. The rat neo-tissue migration was ~70 µm for all the hydrogels. In contrast, the swine neo-tissues penetrated ~70 µm, 130 µm, 150 µm, and 330 µm for CP, GCP, HACP, and CoCP hydrogels, respectively, as evidenced by the Z-stack analysis. Even though neo-tissue formation was evidenced in the CP, GCP, HACP, and CoCP hydrogels, CP, HACP hydrogels were superior to both GCP and CoCP.

### 2.9. Discussion

The natural polysaccharide chitosan has been hailed for cardiac tissue engineering applications owing to its biocompatibility and inherent antimicrobial capacity [17]. Being a polysaccharide, chitosan is superior in myocardial healing and does not provoke aggressive immune reactions [17]. Chitosan retains its crystallinity, rigidity, and stability in the physiological milieu. Additionally, the ample -NH_3_^+^ groups in chitosan interact with anionic polymers, such as gelatin, HA, and PEG, forming polyelectrolyte complexes [18]. Chitosan complexes with biocompatible polymers improves the cardiogenic and biomechanical properties [17,19]. Interestingly, the PEG, upon spontaneously receiving cations including H+, form supra-polyelectrolyte aggregate imparting additional stability and biological features [20]. Hence, the polyelectrolyte complex chemistry of CP hydrogel provides stability and degradability to the hydrogels apart from the inherent biological features of chitosan and PEG. Additionally, the extensive hydrogen bonding stabilize the CP hydrogels and provides opportunities for further complexation with functional groups of complementarily/oppositely charged density. The collagen-derived biomolecule, gelatin, imparts natural adhesion cues for the cell survival and performance [21,22] which inspired us to exploit the biocompatibility of gelatin by modifying CP hydrogel by incorporating gelatin to engineer GCP hydrogels. Similarly, the non-immunogenic cardiac ECM component HA has been successfully exploited in cardiac tissue engineering owing to its active involvement in the biological properties including wound healing, morphogenesis, angiogenesis, and ECM organization [14,23]. Hence, the incorporation of HA in the parent CP hydrogel offers superior benefits in HACP hydrogels and the synergistic effects of gelatin and HA have been exhibited by CoCP.

The ATR-IR analysis revealed the surface decoration of diverse functional groups and chitosan, PEG, HA, and gelatin fragments in the respective hydrogels. The peaks for -OH groups and C-H stretches revealed the formation of CP-based hydrogels consistent with prior reports [24,25]. The N-H bending reflects the polyelectrolyte formation in all four hydrogels [16], which was greater for CP and CoCP, followed by GCP and HACP, as evidenced by the area under the peaks. Also, the IR data suggests the presence of hydrophobic and hydrophilic functional groups/moieties distributed on the surface of CP, GCP, HACP, and CoCP hydrogels. The increased surface -OH groups also facilitate hydrogen bonding, resulting in enhanced water kinetics, cellular adhesion, and biocompatibility [26]. Overall, the surface functional moieties/domains in the CP-based hydrogels favor cell adhesion and survival and guide the seeded cells to deposit ECM, where the surface chemistry allows the integration of neo-ECM with the native cardiac ECM, thereby promoting regeneration [27].

Amphiphilicity promotes biocompatibility, cell migration and protein adsorption in cardiac tissue engineering hydrogels [28,29]. Also, the similar range of ACA and RCA reflects a consistent and biomimetic surface for the cells and biomolecules to interact [30]. Evidently, HA- and gelatin-incorporated hydrogels exhibit a contact angle around 42° [31] as displayed by GCP, HACP and CoCP hydrogels. Additionally, the surfaces with contact angles between 50° and 80° support maximal spreading, growth, and proliferation of multiple cell types, including cardiac progenitor cells and cardiomyocytes (CMs) [32]. Plasma proteins, being the first line of biomolecules encountering the implanted biomaterials and the amphiphilic surfaces, provide cues for interacting with plasma proteins, exhibiting the binding opportunities for both hydrophilic and hydrophobic domains [28,33]. Hence, the amphiphilicity of CP, GCP, HACP, and CoCP hydrogels benefits their proposed cardiac tissue engineering applications.

The water-holding capacity of hydrogels directly impacts biocompatibility, permeability, protein adsorption, and hemocompatibility [34]. Interestingly, superabsorbent hydrogels are characterized by their superior water-holding capacities (EWC > 80%) [24]. When comparing GCP, HACP, and CoCP hydrogels with CP hydrogel, the EWC ~80% in the former hydrogels highlights their superabsorbent nature contributed by the superior water binding capacity of gelatin and HA. Additionally, the increased EWC facilitates the transport and diffusion of oxygen nutrients, metabolites, and exhausts to and from the hydrogel interstices, thus promoting the survival and existence of seeded/recruited cells [35]. A higher EWC and swelling correlate with improved biocompatibility and immunocompatibility [36]. Also, the swelling in hydrogels is a multi-step process involving the diffusion of water molecules, which is a function of the diffusional exponent (*n*). Interestingly, *n* < 0.5 drives swelling-controlled Fickian diffusion, and *n* > 0.5 favors Super Case II Transport (SCIIT) diffusion, which is controlled by network relaxation in hydrogels [1,5]. Our findings revealed that all four, CP, GCP, HACP, and CoCP, hydrogels favor Fickian diffusion, which is crucial for the integration with the native myocardium and for controlling the trafficking of nutrients, biomolecules, and exhausts [37,38]. Additionally, all four hydrogels exhibited a superior TWAS in the order of 10^21^, suggesting their potential to absorb ample water within a relatively short interval, which is responsible for their exceptional water-holding capability [24]. Overall, all four hydrogels bear excellent water profile, which supports cell adhesion, growth, and proliferation and enhances biocompatibility.

Porosity and pore size are critical for the performance of tissue engineering hydrogels as the porosity guides the trafficking of biomolecules, biocompatibility, water kinetics, biodegradation, and cell homing and migration [24,31,35]. Interestingly, a previously prepared chitosan-collagen co-polymer hydrogel displayed pore length similar to GCP, HACP, and CoCP hydrogels, suggesting the impact of chitosan chemistry in inducing porosity [39]. Furthermore, the superabsorbent hydrogels are characterized by superior water content, which in turn, increases porosity. Ideally, cardiac tissue engineering hydrogels require the pore length in the range of 10–100 µm, where pore heterogeneity with small and large pores favor myocardial regeneration [16]. Notably, the larger pores (>80 µm) drive cell migration, angiogenesis, and trafficking of nutrients, metabolites, signaling molecules, and metabolic exhaust, whereas smaller pores (5–20 µm) determine neo-vascularization and cell adhesion [40,41]. Additionally, the porosity and pore size guide the adhered cells to secrete and deposit ECM onto the interstices of the hydrogel, which promote neo-tissue formation [7,8]. Furthermore, imbibing the hydrogels in the physiological medium the pore opening occurs, thereby increasing the room for cellular performance potentiating cardiac regeneration.

Freezing free water and freezing bound water together constitute the total freezable water content (W_f_). The freezable and non-freezing bound water are essential for successful cardiac tissue engineering [25]. Generally, the distribution of hydrophobic/hydrophilic moieties, crosslinking density, and chemical structure play a critical role in the water content and water transition status of hydrogels. Freezable water plays a critical role in trafficking, biocompatibility, adsorption of proteins, and anti-thrombogenicity. In contrast, non-freezable water contributes to pore morphology and mechanical strength [42]. Hence, the key to successful cardiac applications is to balance both water types within the hydrogels. Myocardium is a highly energy-demanding tissue, requiring constant transport of nutrients, metabolites, waste, and ample mechanical strength; thus, hydrogels with appreciable water profile are required [24]. Interestingly, the GCP and HACP hydrogels displayed a balance between freezable and non-freezable water. In contrast, the CP and CoCP hydrogels exhibited minimal non-freezing bound water, suggesting fewer hydrogen bond formation opportunities than the other two counterparts.

The ideal tissue engineering hydrogels support the absorption and release of loaded molecules and cells [24]. We attempted to assess the release profile of the hydrogel using the hydrophilic dye methylene blue and the hydrophobic dye Trypan blue to represent the hydrophilic and hydrophobic biological molecules. Both the dyes exhibited similar release trend supported by the amphiphilic nature of CP, GCP, HACP, and CoCP hydrogels, suggesting the ability of the hydrogels to release loaded molecules, nutrients, and cells. Hence, the CP, GCP, HACP, and CoCP hydrogels establish a proper tissue-hydrogel interface with the native myocardium by facilitating the interactions and transport of biomolecules for cardiac regeneration [43].

Cardiac hydrogels promote the conductance of electrical impulses as in the native myocardium, which is ~1 mS/cm. Practically, the conductive hydrogels inhibit the decoupling of CMs and promote the synchronization of distinct clusters [44] thereby significantly limiting the infarct expansion and CM necrosis due to decoupling [45,46]. The CP, GCP, HACP, and CoCP hydrogels conduct electrical signals. However, the electrical conductivity of the hydrogels was lower than the native myocardium, warranting further modifications, including the addition of conductive polymers, fillers, or ions. Additionally, the biodegradation mechanism by surface erosion displays a relatively constant rate of weight loss, sustaining the release of loaded cells and molecules in a regulated manner. Hence, the stability of CP, GCP, HACP, and CoCP hydrogels in the simulated physiological fluid regulates bulk degradation providing opportunities for protein adsorption and cellular adhesion, proliferation and integration with the host myocardium [24]. Also, the biodegradation of CP, GCP, HACP, and CoCP hydrogels comply with cardiac applications.

The elastic properties imparted by the constant contraction and relaxation cycles of the myocardium warranted to be complied by the mechanical strength of tissue engineering hydrogels to provide ample mechanical support to the damaged areas. An ideal cardiac hydrogel requires the Young’s modulus value to be greater than that of the native myocardium, which in turn has been calculated to be between 20 and 500 kPa. Also, the hydrogels with sufficient mechanical strength significantly inhibit infarct expansion, formation of collagenous scar, and necrosis of the surviving myocardium, and also, cardiac hydrogels capable of providing sufficient mechanical support to the encapsulated cells intensively promote ECM deposition and neo-tissue formation. The GCP, HACP, and CoCP hydrogels exhibited mechanical properties, complying with the native myocardium. In contrast, CP hydrogels, with the lowest concentration of natural materials, exceeded the limit. The increased hydrophilicity imparted by gelatin and HA in GCP, HACP, and CoCP hydrogels possibly reduced the mechanical properties compared to CP hydrogels, representing a strong polyelectrolyte complex. Additionally, the decreased porous nature of CP hydrogels suggests the increased mechanical properties compared to the other three. Translationally, the degradation of hydrogels results in a subsequent decrease in mechanical strength, where the hydrogels with Young’s modulus greater than that of the native tissue exhibit maximal regenerative benefits in cardiac tissue engineering. Overall, all four hydrogels displayed mechanical strength sufficient to support the seeded cells and the diseased myocardium, suggesting ample cross-linking density and regenerative potential.

CP, GCP, HACP, and CoCP hydrogels displayed excellent cytocompatibility with H9c2 cells, as evidenced by the direct contact and MTT viability assays, revealing the non-toxic nature of hydrogel degradation products and leachates. Generally, the hydrogels exhibiting more than 90% cell viability in the test on extract assay have been considered cytocompatible. Interestingly, the hydrogel extracts promoted the H9c2 cell viability over the 90% threshold, suggesting that the degradation products and leachates are non-toxic to the growth and proliferation of cells [42]. Furthermore, the non-toxic nature of the hydrogels is the key to cell adherence, growth, migration, and proliferation [47]. Hence, CP, GCP, HACP, and CoCP hydrogels promise the long-term viability of cells facilitating myocardial regeneration [48]. These findings suggest the non-cytotoxic nature CP, GCP, HACP, and CoCP hydrogels reflecting excellent cytocompatibility and promise in cardiac applications.

The ideal tissue engineering scaffold exhibits minimal hemolytic potential as excessive hemolysis results in immune reactions, anemia, jaundice, and even renal failure and alters the host circulatory system. Interestingly, the hemolytic potential of chitosan-based hydrogels has been reported to be less than 3% [49]. The hemocompatibility of CP, GCP, HACP, and CoCP hydrogels was less than 1%, promising cardiac applications without adverse reactions with the host circulatory system. Additionally, the absence of RBC aggregation by CP, GCP, HACP, and CoCP hydrogels suggests unaltered blood rheology and rouleaux formation upon contact with the circulatory system [24]. These results were consistent with previous reports on PVA-based hydrogels [50]. Furthermore, the interaction of plasma proteins, especially albumin and hydrogel implants, is crucial in determining hemocompatibility. Notably, the amphiphilicity of the hydrogel surface drives plasma protein adsorption, especially that of serum albumin following implantation [24]. Also, the increased freezable water plays a critical role in protein adsorption, where the hydrogels with superior freezing water content display higher protein adsorption [51]. For instance, albumin adsorption onto hydrogel surfaces significantly enhances cell adhesion. Binding to integrin molecules present on cells and to the hydrogel surface facilitates ECM deposition onto hydrogels. Thus, the adsorption of proteins directly enhances the ability of hydrogels to bind to cells and facilitate the formation of neo-tissue [52]. Moreover, the passivation effect of serum albumin enhances biocompatibility, cell adhesion, and hemocompatibility by preventing thrombogenesis, immune reactions, and inflammation [42]. Interestingly, CP, GCP, HACP, and CoCP hydrogels exhibited negligible hemolysis and RBC aggregation and could adsorb serum albumin, reflecting enhanced biocompatibility, hemocompatibility, and anti-thrombogenicity.

A major determinant in the success of cardiac hydrogels is their ability to support the adhesion, growth, and proliferation of cells, resulting in ECM deposition and subsequent integration with the native myocardium. The H9c2 cells grown on CP, GCP, HACP, and CoCP hydrogels displayed the attachment, cytoskeletal spreading, and penetration onto the interstices, suggesting the healthy performance of the cells. The H9c2 cells successfully formed colonies and existed as clusters on CP, GCP, HACP, and CoCP hydrogels, which agrees with previously validated reports [53] suggesting the potential of these hydrogels in initiating tissue formation. Interestingly, all the four hydrogels were capable of neo-tissue formation as tested with rat LV explants which was confirmed with swine myocardial tissue. However, CP and HACP hydrogels were superior to GCP and CoCP. The possible mechanism for the reduced capacity of gelatin-containing hydrogels to support tissue formation is the inherently short degradation time, extreme hydrophilicity, and the involvement of charged domains in poly electrolyte formation limiting the bioavailability of functional cues for cell adhesion [54,55]. Additionally, the cells migrating from the explants and adhering to the surface of hydrogels further accelerate the degradation by breaking hydrogen bonds and activating MMPs prevalent in gelatin-based hydrogels. Nonetheless, the optimal architectural, physiochemical, and biological properties of CP, GCP, HACP, and CoCP hydrogels support neo-tissue organization ex vivo. Translationally, neo-tissue formation reflects the potential of the hydrogels to integrate with the surviving myocardium and to mediate the anterograde and retrograde migration of seeded/host cells post-implantation, which in turn is critical for successful regeneration.

The overall findings from this study demonstrate that the CP, GCP, HACP, and CoCP hydrogels exhibited adequate physiochemical and biological properties, thus benefiting translational cardiac tissue engineering applications. However, further in vivo optimizations in normal and ischemic animal models are warranted to assess the clinical application of these hydrogels. Additionally, the information regarding the cellular/ECM protein deposition/binding to the hydrogels, immuno/histological/biochemical dissection of neo-tissues, and the surgical strategies for implantation warrant careful consideration for translating these hydrogels in clinical arena. Nevertheless, the findings from the present study extend promise for CP, GCP, HACP, and CoCP hydrogel systems in translational regenerative cardiology.

## 3. Conclusions

A panel of four polyelectrolyte complex hydrogels, CP, GCP, HACP, and CoCP, were successfully synthesized. The hydrogels displayed favorable characteristics, such as porosity, water profile and transition status, mechanical strength, hemocompatibility, biocompatibility, enhanced protein adsorption, cell adhesion, growth, proliferation, and neo- tissue formation. Compared to GCP and CoCP, CP and HACP hydrogels were superior owing to cell viability, hemocompatibility, and conductance, resulting in the highest degree of cytoskeletal organization and neo-tissue formation. The excellent physiochemical and biological performance of these hydrogels supported neo-cardiac tissue formation. Overall, the CP, GCP, HACP, and CoCP hydrogel systems promise novel translational opportunities in regenerative cardiology.

## 4. Materials and Methods

### 4.1. Materials

All the chemicals and reagents solutions used for the experiments were of synthetic or analytical grade. Chitosan (medium molecular weight, Cat#448877), glacial acetic acid (Cat#695092), PEG (MW 35,000, Cat#81310), gelatin (gelatin from bovine skin, Cat#G-9382), HA sodium from *Streptococcus equi* (Cat#53747), and dimethyl sulfoxide (Cat#41639) were purchased from Sigma-Aldrich, St. Louis, MO, USA. Dulbecco’s modified Eagle’s medium (DMEM) (Cat#30-2002) was obtained from ATCC, MTT (Cat#M6494), Rhodamine/phalloidin (Cat#R415) was purchased from Invitrogen, 10% neutral buffered formalin (Cat#16004-128) was procured from VWR Labs Private Limited and Pierce Bicinchoninic Acid (BCA) Protein Assay Kit (Cat#23225) was purchased from Thermo Fisher Scientific, Waltham, MA, USA.

### 4.2. Preparation of Chitosan-PEG-Based Polyelectrolyte Hydrogels

Briefly, 3% chitosan in 1% acetic acid and 5% gelatin and 0.1% HA in water were prepared and stored individually at room temperature in airtight containers and were used within 2–3 days for the experiments. The 3% PEG in 3% chitosan solution constituted the parent Chitosan-PEG (CP) blend, which was warmed at 42 °C with constant stirring at 400 rpm for 30 min and casted at 55 °C for 48 h, thus constituting the CP hydrogel. A panel of four hydrogels were prepared from the parent CP blend by incorporating gelatin, and HA in optimal concentrations. Next, 1% gelatin solution was combined with CP blend at 1:2 *v*/*w* ratio, warmed to 42 °C with constant stirring and casted at 55 °C for 48 h to form the gelatin-chitosan-PEG (GCP) hydrogel. HA-chitosan-PEG (HACP) hydrogel was prepared by casting 0.1% HA with CP blend at 1:10 *v*/*w* ratio at 55 °C for 48 h. Finally, 1% gelatin and 0.1% HA were combined (Co) in a 4:1 ratio by stirring for 30 min and mixed with CP blend 1:2 *v*/*w* ratio, heated to 42 °C and casted at 55 °C for 48 h to form the gelatin-HA-chitosan-PEG (CoCP) hydrogel. The hydrogel sheets were then submerged in distilled water, lyophilized, UV sterilized, and stored in aseptic airtight containers for further characterizations and biological studies.

### 4.3. Physiochemical Characterization

#### 4.3.1. Attenuated Total Reflection Infrared Spectroscopy (ATR-IR)

The surface functional groups of lyophilized CP, GCP, HACP, and CoCP hydrogels (*n* = 5, freeze-dried discs) were determined by AT-IR spectral analysis using the IR spectrometer (NICOLET iS50, Thermo Fisher Scientific, Jacksonville, FL, USA). The significant peaks were examined, and the area under the curves was calculated using SPA software associated with the instrument and compared among the groups [56].

#### 4.3.2. Contact Angle

Water-swollen CP, GCP, HACP, and CoCP hydrogels (*n* = 10) of dimension 1 cm × 3 cm with known weight and thickness were used for contact angle analysis. The contact angle was measured using the Wilhelmy method in the KSV Sigma 701 Tensiometer). The advancing contact angle (ACA) and receding contact angle (RCA) in water were automatically recorded by the software associated with the instrument [57].

#### 4.3.3. Water Profiling

Freeze-dried CP, GCP, HACP, and CoCP hydrogels (*n* = 10) were soaked in distilled water and swelling ratio (S), % swelling (%S), equilibrium swelling ratio (E), equilibrium water content (EWC) and total water absorption sites (TWAS) were calculated following our previous protocols [57,58]. The swelling constant and the diffusional exponent were calculated from the intercept and slope of the log(S/E) v. log(time) graph using the equations y = 0.0913 ln(x) + 0.3542, y = −0.079 ln(x) + 0.8751, y = −0.007 ln(x) + 0.8475, and y = −0.017 ln(x) + 1.0641 for CP, GCP, HACP, and CoCP hydrogels, respectively.

#### 4.3.4. Scanning Electron Microscopy (SEM)

The microarchitecture and morphology of the cross-sectional area of freeze-dried hydrogels were characterized using scanning electron microscopy (Nova NanoSEM 450; Fei Co., Hillsboro, OR, USA). Samples were sputter-coated with a Pd/Pt layer for 60 s at 20 mA using a sputter coater (108 Auto Sputter Coater; Ted Pella, Inc., Redding, CA, USA) before being imaged. An accelerating voltage of 20 kV and a working distance of 5 mm were used, along with a spot size of 3.0. The images were processed using ImageJ(1.46) software to quantify pore size and aspect ratios of the pores, following our previously validated protocols [59,60].

#### 4.3.5. Thermal Evaluation

The thermal properties of water swollen hydrogels were assessed using differential scanning calorimetry (DSC214; Netzsch, Germany) to determine water transition states following previous protocols. Water-swollen hydrogels were cooled from 0 °C to −40 °C, and then heated to 100 °C at a rate of 5 K/min in a nitrogen (N2) atmosphere. The heating and cooling curves were recorded. The freezing water content (W_f_), freezing free water (W_ff_), freezing bound water (slightly structured) (W_fb_), and non-freezing bound water (W_nb_) were calculated following our previously published reports [57,60].

### 4.4. Release Kinetics

CP, GCP, HACP, and CoCP hydrogels (*n* = 4) of known dry weights were submerged in the hydrophobic dye trypan blue (5 mg%), and the hydrophilic dye methylene blue (5 mg%), for 24 h. The unbound dyes were washed in PBS, and the burst release profiles of the dyes from each hydrogel were observed every hour for 8 h, followed by the sustained release profiles once in 2 days for 28 days by extracting PBS. The concentration of released dyes was calculated from the absorption values measured using a plate reader. A standard curve was plotted, and the percentage release of the dyes was determined from our already established protocols [56].

### 4.5. Electrical Conductivity

Water-swollen CP, GCP, HACP, and CoCP hydrogel discs (*n* = 5) were placed on a non-conductive surface, and resistance was assessed using a multimeter. Conductance was then calculated from the resistance using the following equation, where *G* is conductance, and *R* is resistance.
$$G=\frac{1}{R}$$

### 4.6. Biodegradation

Degradation of the CP, GCP, HACP, and CoCP hydrogels (*n* = 5) was assessed by aging in PBS (pH = 7.33) at 37 °C. Weight loss (difference wet weight/initial wet weight × 100) was assessed weekly for seven weeks and the changes in pH were measured weekly to determine the acidity/basicity of degradation products [61].

### 4.7. Mechanical Characterization

The mechanical strength of the water swollen CP, GCP, HACP, and CoCP hydrogels was (*n* = 6) was measured in the universal testing machine (Instron 5943 dual-column testing system, Norwood, MA, USA) equipped with a 10 N load cell (Instron). The tensile strength was determined using a 0.005 N preload with a 10 mm/min crosshead speed. Tensile stress at failure, load at failure, and Young’s modulus were calculated automatically by the program associated with the instrument [62].

### 4.8. Cytocompatibility

#### 4.8.1. Cell Culture and Maintenance

Rat cardiomyoblasts, H9c2 cells (Cat# CLR1446; ATCC), were used for cytocompatibility assessment, and the cells were maintained in high glucose DMEM with 10% FBS under standard culture conditions (5% CO_2_, 37 °C, and antibiotics). H9c2 from passages 2–6 was used for the studies.

#### 4.8.2. Direct Contact Assay

Lyophilized hydrogels (*n* = 5) were swollen in DMEM with 10% FBS for 24 h, placed atop a sub-confluent layer of H9c2 cells and incubated for 24 h. The cells at the interface of the hydrogels were imaged to assess the changes in morphological alterations using a phase-contrast microscope (Olympus CKx41). The overall survival of the H9c2 cells was quantified by MTT cell viability assay as previously validated [63]. H9c2 cells cultured under the same conditions without hydrogels served as control.

#### 4.8.3. Test on Extract

DMEM with 10% FBS was used to extract the lyophilized CP, GCP, HACP, and CoCP hydrogels (*n* = 4) for 48 h and to culture H9c2 cells. The cell viability was quantified after three days using the MTT assay. H9c2 cells cultured under the same conditions without hydrogels served as control.

### 4.9. Hemocompatibility

#### 4.9.1. Hemolysis Assay Red Blood Cell Aggregation

Lyophilized CP, GCP, HACP, and CoCP hydrogels (*n* = 3) were soaked in 0.9% saline solution for 48 h. Briefly, 5 mL blood for the hemocompatibility was collected from Yucatan micro swine (from a different study) immediately prior to sacrifice (*Sus scorfa*; Sinclair BioResources) with approval from the IACUC at Western University of Health Sciences (R21IACUC012). The plasma was separated, and the RBCs were washed and diluted 10 times with a 0.9% saline solution. Then, 0.5 mL diluted RBC was mixed gently with 0.5 mL saline extract of each hydrogel, incubated at 37 °C for 30 min, centrifuged, and the absorbance of the supernatant was read at 540 nm. RBC in 0.5 mL 0.9% saline solution served as the negative control, and 0.5 mL RBC in 0.5 mL deionized water served as the positive control. The percentage of hemolysis was calculated using previously reported protocols [56]. The smears of the above samples were examined under a brightfield microscope (Olympus CKx41) to assess the morphology of red blood cells and rouleau formations [56].

#### 4.9.2. Protein Adsorption

Lyophilized CP, GCP, HACP, and CoCP hydrogels (*n* = 5) were incubated in 10% plasma isolated from Yucatán micro swine (*Sus scorfa*, Sinclair Bioresources) at 37 °C overnight. The loosely bound proteins adsorbed onto the hydrogels were washed with sterile-filtered PBS, and the adsorbed proteins were extracted in 500 µL PBS by vigorous vortexing and centrifuged to remove the debris. The extracted protein content was quantified using the BCA assay [16]. Also, the level of serum albumin adsorbed on CP, GCP, HACP, and CoCP hydrogels was assessed using sodium dodecyl sulfate-polyacrylamide gel electrophoresis (SDS-PAGE) using bovine serum albumin (BSA) as a reference. Relative protein concentration was determined by calculating band intensity using ImageJ software [56].

### 4.10. Biological Performance

#### 4.10.1. Cell Spreading and Penetration

H9c2 cells were seeded onto the CP, GCP, HACP, and CoCP DMEM-swollen hydrogels (*n* = 3) and allowed to grow under standard culture conditions for six days. The hydrogels were fixed in formalin, blocked with 1% horse-blocking serum (HBS), stained with rhodamine phalloidin for 2 h, and imaged using a fluorescent microscope (Leica Thunder, Wetzlar, Germany). Z-stack images were captured, 3D rendering was performed, and Z-stack depth was measured for each sample to assess the penetration [56]. Cells grown on glass coverslips were used as controls.

#### 4.10.2. Ex Vivo Explant Culture and Neo-Tissue Formation

Left ventricular tissue sections (~50 mg weight) were harvested from cadaver rats and swine in serum-free DMEM and were allowed to grow on CP, GCP, HACP, and CoCP hydrogels (*n* = 3) equilibrated in DMEM with 20% FBS. After 1 h of attachment, additional media was added without disturbing the tissue and allowed to grow for one week. Then, the hydrogels were fixed after removing the tissues, and rhodamine staining was performed to observe the neo-tissue organization in the hydrogels, as mentioned above.

#### 4.10.3. Statistical Analysis

The results of all experiments were expressed as mean ± SEM, and all the experiments were run in at least three or more replicates, excluding SEM and DSC. ImageJ software was used for pore measurements, DSC area calculations, and SDS-PAGE band quantification. The statistical significance for all the experimental data was determined by one-way ANOVA with a two-stage linear step-up procedure with Tukey’s multiple comparisons. However, the statistical significance for contact angle measurement was determined by an unpaired “*t*” test for comparing two groups. Statistical significance of *p* < 0.05 was set for all experiments, and GraphPad Prism 9.4.1 (681) program was employed for the analysis.

## Figures and Tables

**Figure 1 gels-10-00046-f001:**
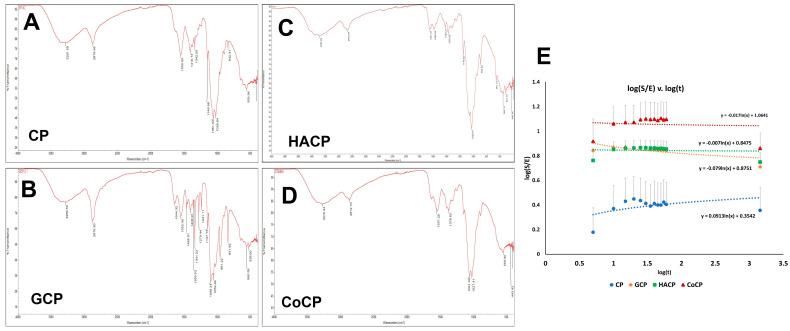
ATR-IR spectrum for CP (**A**), GCP (**B**), HACP (**C**), and CoCP (**D**) hydrogels; the X-axis shows % transmittance, and the Y-axis shows wavenumber (cm^−1^). (**E**) Determination of diffusional exponent (n) from the slope and swelling constant (k) from the Y-intercept.

**Figure 2 gels-10-00046-f002:**
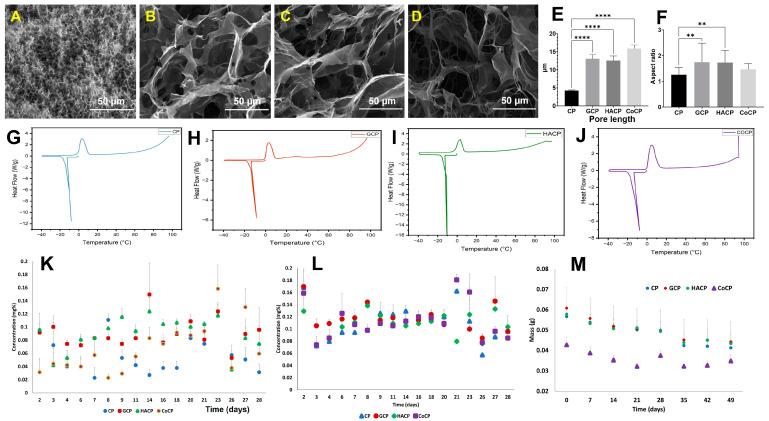
Representative SEM images demonstrating cross-sectional morphology of CP (**A**), GCP (**B**), HACP (**C**), and CoCP (**D**) hydrogels. Bar diagram depicting hydrogel pore length (**E**) and pore aspect ratio (**F**). DSC thermograms depicting the melting and crystallization of water in CP (**G**), GCP (**H**), HACP (**I**), and CoCP (**J**) hydrogels. Long-term dye dissociation of Methylene Blue (**K**) and Trypan Blue (**L**) from CP, GCP, HACP, and CoCP hydrogels. (**M**) Alterations in the weight of CP, GCP, HACP, and CoCP hydrogels upon aging in PBS. (** *p* < 0.01, **** *p* < 0.0001 and unlabeled parameters are nonsignificant).

**Figure 3 gels-10-00046-f003:**
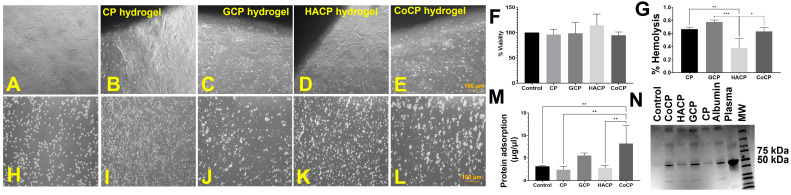
(**A**) Direct contact assay control with no hydrogels. Representative images displaying a lack of morphological changes in H9c2 cardiomyoblasts when in direct contact with CP (**B**), GCP (**C**), HACP (**D**), and CoCP (**E**) hydrogels. (**F**) Cell viability following the growth of H9c2 cells in hydrogel extracts. (**G**) Hemolytic potential of the hydrogels. Representative images of RBC aggregation assay showing negative control (**H**), CP (**I**), GCP (**J**), HACP (**K**), and CoCP (**L**) hydrogels. Quantification of total plasma protein adsorption by the hydrogels (**M**) and the SDS-PAGE (**N**) showing the adsorption of albumin (* *p* < 0.05, ** *p* < 0.01, *** *p* < 0.001, unlabeled parameters are not significant).

**Figure 4 gels-10-00046-f004:**
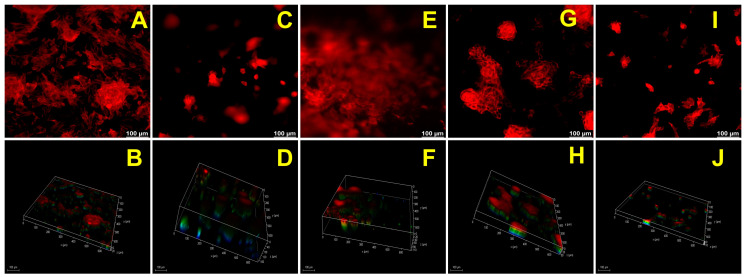
Rhodamine phalloidin-stained H9c2 cells seeded atop control (**A**), CP hydrogel (**C**), GCP hydrogel (**E**), HACP hydrogel (**G**), and CoCP hydrogel (**I**). Twenty consecutive Z-stack images and 3D rendering showing control (**B**), CP hydrogel (**D**), GCP hydrogel (**F**), HACP hydrogel (**H**), and CoCP hydrogel (**J**). Images were acquired at 20× magnification; scale bar: 100 µm.

**Figure 5 gels-10-00046-f005:**
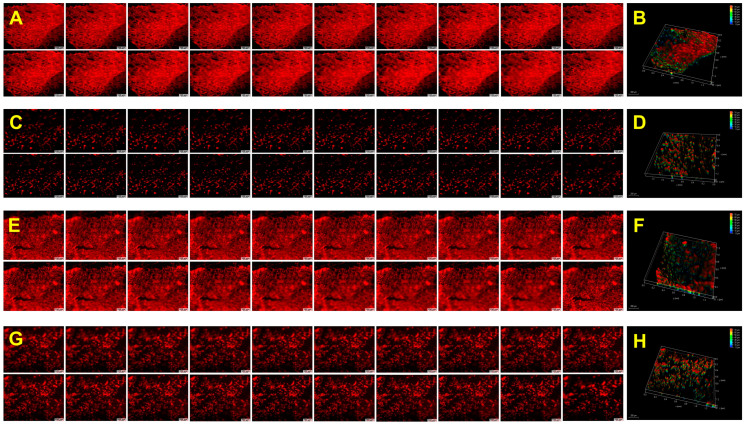
Neo-tissue formation by rat LV explants: 20 consecutive z-stack images and 3D rendering images of rhodamine phalloidin for CP hydrogel (**A**,**B**), GCP hydrogel (**C**,**D**), HACP hydrogel (**E**,**F**), and CoCP hydrogel (**G**,**H**), respectively, after a week of seeding. Images were acquired at 10× magnification; scale bar: 100 µm.

**Figure 6 gels-10-00046-f006:**
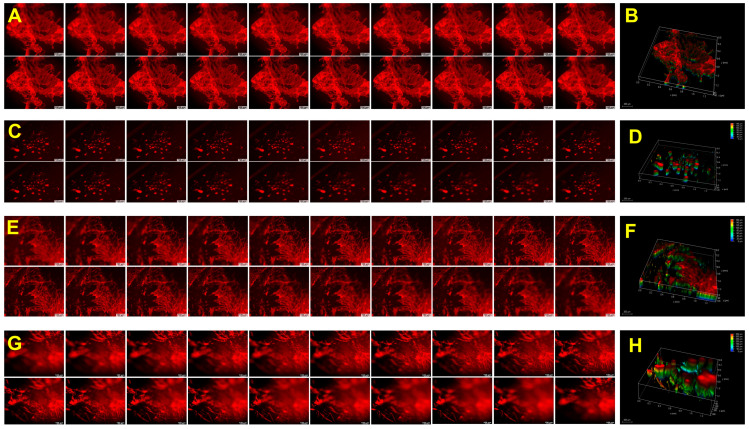
Neo-tissue formation by swine LV explants: 20 consecutive z-stack images and 3D rendering images of rhodamine phalloidin for CP hydrogel (**A**,**B**), GCP hydrogel (**C**,**D**), HACP hydrogel (**E**,**F**), and CoCP hydrogel (**G**,**H**), respectively, after a week of seeding. Images were acquired at 10× magnification; scale bar: 100 µm.

**Table 1 gels-10-00046-t001:** Characteristic features of CP, GCP, HACP and CoCP hydrogels.

Parameters	CP	GCP	HACP	CoCP
ACA (°) (*n* = 10)	29.38 ± 2.57	43.70 ± 4.60	44.401 ± 3.90	47.21 ± 7.20
RCA (°) (*n* = 10)	31.83 ± 2.42	46.55 ± 5.99	42.38 ± 7.10	49.05 ± 5.37
Swelling Ratio (S) (*n* = 10)	2.47 ± 0.12	4.48 ± 0.66	5.51 ± 1.14	9.27 ± 2.46
Equilibrium Swelling Ratio (E) (*n* = 10)	0.71 ± 0.01	0.79 ± 0.02	0.80 ± 0.03	0.81 ± 0.05
% Swelling (%) (*n* = 10)	247.3 ± 11.8	447.8 ± 65.62	551.2 ± 113.6	926.8 ± 245.9
Equilibrium Water Content (EWC) (%) (*n* = 10)	70.92 ± 0.94	79.12 ± 2.43	79.56 ± 3.30	80.65 ± 4.54
Total Water Absorption Sites (TWAS) (*n* = 10)	1.129 × 10^21^ ± 3.67 × 10^19^	1.520 × 10^21^ ± 2.85 × 10^20^	1.679 × 10^21^ ± 2.20 × 10^20^	1.433 × 10^21^ ± 4.53 × 10^20^
Diffusional Exponent (n) (*n* = 10)	0.0913	−0.079	−0.007	−0.017
Diffusion Constant (k) (*n* = 10)	0.3542	0.8751	0.8475	1.0641
Pore Length (µm) (*n* > 20)	4.241 ± 0.181	13.089 ± 1.130	12.532 ± 1.301	15.881 ± 1.104
Aspect Ratio (*n* > 20)	1.256 ± 0.054	1.746 ± 0.165	1.728 ± 0.106	1.461 ± 0.047
Enthalpy of Melting Endotherm (J/g)	237.3	122.0	188.2	277.0
Freezing Water Content (%)	69.58	34.28	52.93	71.92
Non-freezing Water Content (%)	1.34	44.91	26.63	8.74
Conductance (µS/cm) (*n* = 5)	0.28 ± 0.01	0.69 ± 0.07	2.60 ± 1.23	0.44 ± 0.02
Young’s Modulus (kPa) (*n* = 6)	1141.0 ± 241.1	179.4 ± 51.5	368.4 ± 95.4	131.7 ± 16.7
Tensile Stress at Failure (kPa) (*n* = 6)	401.9 ± 54.2	39.22 ± 2.12	96.41 ± 18.81	47.22 ± 5.71
Load at Failure (N) (*n* = 6)	5.63 ± 0.75	0.94 ± 0.03	1.74 ± 0.31	1.34 ± 0.12
Direct Contact—MTT assay (%Viability) (*n* = 5)	71.88 ± 3.26	82.84 ± 2.82	92.44 ± 0.79	74.09 ± 1.43
Test on Extract—MTT assay (% Cell) (*n* = 4)	96.1 ± 5.2	98.8 ± 10.4	114.4 ± 11.1	94.9 ± 3.2
% Hemolysis (*n* = 3)	0.66 ± 0.02	0.76 ± 0.02	0.27 ± 0.08	0.63 ± 0.04
Absorption of Total Plasma Protein (µg/µL) (*n* = 5)	1.67 ± 0.51	5.48 ± 0.26	2.78 ± 0.26	8.19 ± 1.8

## Data Availability

All data and materials are available on request from the corresponding author. The data are not publicly available due to ongoing research using a part of the data.

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
