# Peer review of "Chitosan–Polyethylene Glycol Inspired Polyelectrolyte Complex Hydrogel Templates Favoring NEO-Tissue Formation for Cardiac Tissue Engineering"

_gels, 2024, doi:10.3390/gels10010046_

Round 1

Reviewer 1 Report

Comments and Suggestions for Authors

Chitosan-polyethylene glycol inspired polyelectrolyte complex 2 hydrogel templates favoring neo-tissue formation for cardiac 3 tissue engineering.

Major revision  

In general, the authors did not clearly emphasized the strengths of their study/methods in abstract and conclusion?

In the abstract

-          Symbols such as (CP, GCP, etc…) should be expanded

-          Include some numerical results

In the Introduction

-          Different font style '' synthesize novel 77 polyelectrolyte hydrogels from chitosan, PEG, gelatin, and HA, characterizing the 78 properties and performance, and evaluating the potential in neo-tissue formation 79 for cardiac tissue engineering applications. ''

-          The utilized characterization techniques to fulfil the study aim should be included at the end of introduction.

In materials and methods

-          What is VWR? Expand.

-          '' 3% chitosan in 1% acetic acid, and 5% gelatin and 0.1% HA in water were prepared 93 and stored at room temperature in airtight containers and were used within 2-3 days for 94 the experiments'', are these solutions in mixture or in separate forms ? Please clarify and what are the preparation conditions of these solutions?

-          For all the utilized characterization techniques the authors should include the instrument specifications including origin country and so on ---.

-          Samples preparation before measurements should be stated in the methodology section for each characterization technique.

-          '' The thermal properties of hydrogels were assessed using differential scanning calorimetry (DSC214; Netzsch) to determine water transition states following previous protocols.''  Add the protocol reference (s).

-          Which instrument was used to measure the trypan and methylene blue release?

-          Include the equation used to calculate the biodegradation.

-          The approval No. of ethics should be included in the cells studies.

In the results

-          The text in Figure 1(A-D) is not visible

-          Where is Table 1?

-          How the authors be sure that the pores possess spherical shape?

-          In Figure 2(G-J) include and indicate the different types of water on their crossponding peaks.

-          In this sentence '' The DSC thermograms for CP, GCP, HACP and CoCP hydrogels displayed corresponding endothermic and endothermic respectively reflecting the melting of frozen water and crystallization of freezing water (Figure 2G-2J)'', is it should be exothermic and endothermic?

-          All the modified gels showed lower mechanical properties compared to the CP gel, what is the authors' explanation for this failure?  

-          Add scale bare in all the images of figures 4, 5 and 6.

Author Response

Reviewer: #1

Concern #1: In general, the authors did not clearly emphasized the strengths of their study/methods in abstract and conclusion?

Response: Thank you for your valuable suggestion! The following information has been included in the abstract and conclusion.

‘The physiochemical and biological performance of these hydrogels supported neo-cardiac tissue formation.’

In the abstract

Concern #2: Symbols such as (CP, GCP, etc…) should be expanded.

Response: Thank you for your comment! The abstract has been modified as suggested.

Concern #3: Include some numerical results

Response: Thank you for your comment! The abstract has been modified as suggested.

In the Introduction

Concern #4: Different font style '' synthesize novel 77 polyelectrolyte hydrogels from chitosan, PEG, gelatin, and HA, characterizing the 78 properties and performance, and evaluating the potential in neo-tissue formation 79 for cardiac tissue engineering applications. ''

Response: Thank you for your input! The font size has been corrected as suggested.

Concern #5: The utilized characterization techniques to fulfil the study aim should be included at the end of introduction.

Response: Thank you for your suggestion! The following information has been included in the introduction.

‘The hydrogels were tested for physiochemical, mechanical and thermal characteristics and biological performance to assess their ability for neotissue formation.’

In materials and methods

Concern #6: What is VWR? Expand.

Response: Thank you for your comment! The methodology has been modified as suggested.

Concern #7: '' 3% chitosan in 1% acetic acid, and 5% gelatin and 0.1% HA in water were prepared 93 and stored at room temperature in airtight containers and were used within 2-3 days for 94 the experiments'', are these solutions in mixture or in separate forms ? Please clarify and what are the preparation conditions of these solutions?

Response: Thank you for your suggestion! The methodology has been modified as suggested.

Concern #8: For all the utilized characterization techniques the authors should include the instrument specifications including origin country and so on ---.

Response: Thank you for your suggestion! The methodology has been modified as suggested.

Concern #9: Samples preparation before measurements should be stated in the methodology section for each characterization technique.

Response: Thank you for your suggestion! The methodology has been modified as suggested.

Concern #10: '' The thermal properties of hydrogels were assessed using differential scanning calorimetry (DSC214; Netzsch) to determine water transition states following previous protocols.''  Add the protocol reference (s).

Response: Thank you for your suggestion! The reference #20 has already been included.

Concern #11: Which instrument was used to measure the trypan and methylene blue release?

Response: Thank you for your suggestion! The following information has been included in the revised manuscript.

‘The concentration of released dyes was calculated from the absorption values measured using a plate reader. A standard curve was ploted and the percentage release of the dyes was determined from our already established protocols [16].’

Concern #12: Include the equation used to calculate the biodegradation.

Response: Thank you for your suggestion! The methodology has been modified as suggested.

Concern #13: The approval No. of ethics should be included in the cells studies.

Response: Thank you for your comment! The methodology has been modified as suggested.

In the results

Concern #14: The text in Figure 1(A-D) is not visible

Response: Thank you for your comment! Higher resolution figure has been included in the revised manuscript.

Concern #15: Where is Table 1?

Response: Thank you for your comment! Table has been included in the revised manuscript.

Concern #16: How the authors be sure that the pores possess spherical shape?

Response: Thank you for your suggestion! The aspect ratio is the measure of circularity and the aspect ratio >1 signifies oval shape. This has been corrected in the revised manuscript.

Concern #17: In Figure 2(G-J) include and indicate the different types of water on their crossponding peaks.

Response: Thank you for your comment! The peaks in the thermogram reflect the enthalpy of melting endotherm/exotherm. The endotherm/exotherm values are utilized for calculating different types of water.

Concern #18: In this sentence '' The DSC thermograms for CP, GCP, HACP and CoCP hydrogels displayed corresponding endothermic and endothermic respectively reflecting the melting of frozen water and crystallization of freezing water (Figure 2G-2J)'', is it should be exothermic and endothermic?

Response: Thank you for your comment! Table has been included in the revised manuscript.

Concern #19: All the modified gels showed lower mechanical properties compared to the CP gel, what is the authors' explanation for this failure?  

Response: Thank you for your critical comment! The following information has been included in the discussion section of revised manuscript.

‘The increased hydrophilicity imparted by gelatin and HA in GCP, HACP, and CoCP hydrogels possibly reduced the mechanical properties compared to CP hydrogels, representing a strong polyelectrolyte complex. Additionally, the decreased porous nature of CP hydrogels suggests the increased mechanical properties compared to the other three.’

Concern #20: Add scale bare in all the images of figures 4, 5 and 6.

Response: Thank you for your suggestion! The scale bars have already been included in the figures.

Reviewer 2 Report

Comments and Suggestions for Authors

The reviewed manuscript is devoted to an urgent problem; the research was carried out at a high scientific level using methods adequate to the tasks set. At the same time, there is a clear disproportion between the results, the chemical and biological parts. If biocompatibility and hemocompatibility have been reliably established and the prospects for using the resulting hydrogel are not in doubt, then the issue of the mechanism of gel formation has not been considered in such detail. Therefore, the following questions and recommendations arise:

1) Polyethylene glycol is not a polyelectrolyte, so is it correct to call the resulting gels polyelectrolyte complexes? It seems that the formation of interpolymer hydrogen bonds may play a significant role here.

2) Give the scheme of the quasi-reaction leading to the formation of an interpolymer complex. This will help readers understand the proposed mechanism of hydrogel formation.

3) Compare the results obtained on hemocompatibility with those obtained on other polymers, for example, on branched PVA (doi: 10.1134/S1070427216080115). Show the benefits of the gels described in the manuscript.

Author Response

Reviewer 2

The reviewed manuscript is devoted to an urgent problem; the research was carried out at a high scientific level using methods adequate to the tasks set. At the same time, there is a clear disproportion between the results, the chemical and biological parts. If biocompatibility and hemocompatibility have been reliably established and the prospects for using the resulting hydrogel are not in doubt, then the issue of the mechanism of gel formation has not been considered in such detail. Therefore, the following questions and recommendations arise:

Concern #1: Polyethylene glycol is not a polyelectrolyte, so is it correct to call the resulting gels polyelectrolyte complexes? It seems that the formation of interpolymer hydrogen bonds may play a significant role here.

Response: Thank you for your critical comment! We agree that H bonding plays a significant role in stabilizing the hydrogels. The following justification has been included in the revised manuscript.

‘The ample -NH3+ groups in chitosan interact with anionic polymers, such as gelatin, HA, and PEG, forming polyelectrolyte complexes [26]. Complexing chitosan with biocompatible polymers improves the cardiogenic and biomechanical properties [25] [27]. Interestingly, the PEG upon receiving cations including H+ spontaneously form supra-polyelectrolyte aggregate imparting additional stability and biological features [28]. Hence, the polyelectrolyte complex chemistry of CP hydrogel provides stability and degradability to the hydrogels apart from the inherent biological features of chitosan and PEG. Additionally, the extensive hydrogen bonding stabilize the CP hydrogels and provides opportunities for further complexation with functional groups of complementarily/oppositely charged density.’

Concern #2: Give the scheme of the quasi-reaction leading to the formation of an interpolymer complex. This will help readers understand the proposed mechanism of hydrogel formation.

Response: Thank you for your comment! Figure for hydrogel formation schema has been included in the revised manuscript as supplementary file.

Concern #3: Compare the results obtained on hemocompatibility with those obtained on other polymers, for example, on branched PVA (doi: 10.1134/S1070427216080115). Show the benefits of the gels described in the manuscript.

Response: Thank you for your suggestion! The discussion section has been modified as below by including the recommended reference.

‘The ideal tissue engineering scaffold exhibits minimal hemolytic potential as excessive hemolysis results in immune reactions, anemia, jaundice, and even renal failure and alters the host circulatory system. Interestingly, the hemolytic potential of chitosan-based hydrogels has been reported to be less than 3% [58]. The hemocompatibility of CP, GCP, HACP, and CoCP hydrogels was less than 1%, promising cardiac applications without adverse reactions with the host circulatory system. Additionally, the absence of RBC aggregation by CP, GCP, HACP, and CoCP hydrogels suggests unaltered blood rheology and rouleaux formation upon contact with the circulatory system [59]. These results were consistent with previous reports on PVA based hydrogels [60].’

Round 2

Reviewer 1 Report

Comments and Suggestions for Authors

Authors have covered most of comments/suggestions and it is now accepted for publication 

Reviewer 2 Report

Comments and Suggestions for Authors

The authors reviewed the article as recommended. I recommend accepting the manuscript.